# HCV Activates Somatic L1 Retrotransposition—A Potential Hepatocarcinogenesis Pathway

**DOI:** 10.3390/cancers13205079

**Published:** 2021-10-11

**Authors:** Praveen D. Sudhindar, Daniel Wainwright, Santu Saha, Rachel Howarth, Misti McCain, Yvonne Bury, Sweta S. Saha, Stuart McPherson, Helen Reeves, Arvind H. Patel, Geoffrey J. Faulkner, John Lunec, Ruchi Shukla

**Affiliations:** 1Newcastle University Centre for Cancer, Biosciences Institute, Faculty of Medical Sciences, The Medical School, Newcastle University, Newcastle upon Tyne NE2 4HH, UK; p.dhondurao-sudhindar2@newcastle.ac.uk (P.D.S.); Daniel.Wainwright@newcastle.ac.uk (D.W.); Rachel.Howarth@newcastle.ac.uk (R.H.); john.lunec@newcastle.ac.uk (J.L.); 2Newcastle University Centre for Cancer, Translational and Clinical Research Institute, Faculty of Medical Sciences, The Medical School, Newcastle University, Newcastle upon Tyne NE2 4HH, UK; Santu.Saha@newcastle.ac.uk (S.S.); Misti.McCain@newcastle.ac.uk (M.M.); Sweta.Sharma-Saha@newcastle.ac.uk (S.S.S.); Helen.Reeves@newcastle.ac.uk (H.R.); 3Department of Cellular Pathology, Royal Victoria Infirmary, Newcastle upon Tyne Hospitals NHS Foundation Trust, Newcastle upon Tyne NE1 4LP, UK; yvonne.bury@nhs.net; 4The Liver Unit, Freeman Hospital, Newcastle upon Tyne Hospitals NHS Foundation Trust, Heaton NE7 7DN, UK; stuart.mcpherson2@nhs.net; 5MRC-University of Glasgow Centre for Virus Research, University of Glasgow, Glasgow G61 1QH, UK; arvind.patel@glasgow.ac.uk; 6Mater Research Institute, University of Queensland, Woolloongabba, QLD 4102, Australia; geoffrey.faulkner@mater.uq.edu.au; 7Queensland Brain Institute, University of Queensland, Brisbane, QLD 4072, Australia

**Keywords:** L1, HCV, retrotransposition, DNA damage, hepatocellular carcinoma

## Abstract

**Simple Summary:**

Chronic hepatitis C virus (HCV) infection is a common cause of liver cancer in the developed world. Although anti-viral treatment can cure HCV infection, the risk of cancer development remains in individuals with consequent advanced fibrosis or cirrhosis. In this study, we show that HCV can influence transposable DNA elements called L1 retrotransposons. These are mobile genetic elements that can negatively alter the host genome, potentially promoting cancer development. L1 elements are known to be activated in several cancer types, including liver cancer. Hence, we suggest a novel pathway involved in liver cancer development in patients with chronic HCV, including those with active infection as well as after viral clearance.

**Abstract:**

Hepatitis C virus (HCV) is a common cause of hepatocellular carcinoma (HCC). The activation and mutagenic consequences of L1 retrotransposons in virus-associated-HCC have been documented. However, the direct influence of HCV upon L1 elements is unclear, and is the focus of the present study. L1 transcript expression was evaluated in a publicly available liver tissue RNA-seq dataset from patients with chronic HCV hepatitis (CHC), as well as healthy controls. L1 transcript expression was significantly higher in CHC than in controls. L1orf1p (a L1 encoded protein) expression was observed in six out of 11 CHC livers by immunohistochemistry. To evaluate the influence of HCV on retrotransposition efficiency, in vitro engineered-L1 retrotransposition assays were employed in Huh7 cells in the presence and absence of an HCV replicon. An increased retrotransposition rate was observed in the presence of replicating HCV RNA, and persisted in cells after viral clearance due to sofosbuvir (PSI7977) treatment. Increased retrotransposition could be due to dysregulation of the DNA-damage repair response, including homologous recombination, due to HCV infection. Altogether these data suggest that L1 expression can be activated before oncogenic transformation in CHC patients, with HCV-upregulated retrotransposition potentially contributing to HCC genomic instability and a risk of transformation that persists post-viral clearance.

## 1. Introduction

Hepatocellular carcinoma (HCC) is the sixth most common type of cancer and fourth most frequent cause of cancer-related deaths worldwide [1]. One of the most important aetiological factors associated with HCC development is chronic hepatitis B or C virus (HBV or HCV) infection [2]. A major advance in the field has been the introduction of direct-acting antivirals (DAAs) targeting HCV infection, which induce very high rates of sustained viral clearance, and now the majority of the patients treated with DAAs are cured of HCV infection [3]. However, while successful treatment of HCV reduces the risk of developing HCC, this risk is not completely eliminated, especially amongst patients who have advanced fibrosis or cirrhosis [4]. A large proportion of patients presenting with HCC have incurable disease at presentation, due to late detection when curative treatments (transplantation, resection, ablation) cannot be offered. This negatively impacts survival. Early detection of HCC greatly increases the likelihood of curative therapies being offered. This presents an unmet need—a better understanding of the HCC risk in HCV patients is needed in order to develop tools for improved HCC early detection and outcomes [5,6]. We believe that understanding the molecular mechanisms that lead to hepatocarcinogenesis in HCV will help to achieve this.

The molecular mechanism for the development of HCV-associated HCC remains unknown and is likely to be multifactorial, involving multiple molecular pathways. Upon HCV infection, the virus modulates the host cells for its own survival and replication. The HCV genome is directly translated at the rough endoplasmic reticulum (ER) as a single polyprotein precursor that is eventually cleaved by cellular and viral proteases into ten mature products. These virus-encoded proteins then further participate in the process of viral replication and assembly [7]. Overall, the HCV replication process induces oxidative and ER stress in the liver cells, promoting hepatocarcinogenesis [8,9,10]. Moreover, HCV infection has been demonstrated to induce autophagy [11], the innate immune response [12] and to impair DNA damage repair (DDR) pathways [13,14,15]. Hence, HCV-encoded proteins interact with various host proteins and dysregulate various pathways, contributing towards hepatocarcinogenesis. Several of these changes are epigenetic in nature [16,17], a number of which have been shown to persist even after HCV infection clearance by DAA treatment [18,19].

Okamoto et al. previously demonstrated global DNA hypomethylation (indicated by the L1 promoter) as a consequence of HBV and HCV infection in a humanised mouse model of hepatitis virus infection [20]. Furthermore, Shukla et al. demonstrated L1 (an autonomous mobile genetic element, or retrotransposon) activation and its promotion of oncogenic signalling pathways in HBV- and HCV-related HCC [21]. Incidentally, active L1 retrotransposition in virus (HCV or HBV)-associated HCC was also reported by the Pan-Cancer Whole-Genome Analysis (PCWGA) consortium study, which included HCC cases [22]. Active L1 retrotransposition has also been observed in cases of alcohol-related-liver-disease-associated HCC [22,23]. However, the rate of somatic L1 retrotransposition was lower in alcohol-related cases (~14%; five out of 37 individuals) compared to virus-associated HCC (~32%; 97 out of 306 individuals) [21,22,23]. Somatic retrotransposition in HCC has been hypothesised to be an early event in hepatocarcinogenesis, at least in the presence of HBV-HCC, where it has been detected in the non-tumour liver tissue of one HBV-HCC case [21]. Recently, HCV infection has also been associated with the loss of DNA methylation in specific repeat elements (L1s and Alu), showing a stepwise hypomethylation from normal liver to HCV-cirrhosis to HCV-HCC [24]. It is well established that L1 elements activated in various cancer types are involved in somatic structural variations, leading to cancer evolution, with additional roles in human cancer development also being implicated [22,25].

Hence, we hypothesise that L1s are activated in CHC patients, due to virus-assisted epigenetic remodeling and suppression of host defense factors. We aimed to confirm this in human samples and to evaluate the influence of HCV on the process of active retrotransposition using in vitro models. Once activated, L1s may promote genomic instability, contributing towards cancer development even after HCV clearance. These mechanistic understandings may help with future therapeutic or cancer-preventative approaches for CHC patients.

## 2. Materials and Methods

### 2.1. Cell Lines and Chemical Inhibitors

The parental line Huh7 (a kind gift from Jean Dubuisson, Institut Pasteur de Lille, Lille Cedex, France) and its derivative line Huh7-J17 were routinely maintained in RPMI 1640 media (Sigma, St. Louis, MO, USA, R5886) with 10% fetal bovine serum, 1% penicillin and streptomycin and 1% L-glutamine. Huh7-J17 cells stably express an HCV sub-genomic replicon encoding a firefly luciferase reporter and a puromycin resistance marker (separated by the foot-and-mouth-disease virus (FMDV) 2a self-cleavage site) in the genotype 2a HCV strain JFH1 ΔE1E2 background, as described previously [26,27]. Hence, 2 µg/mL puromycin (Sigma P8833) was added into the media of Huh7-J17 cells to maintain the selection of HCV replicon-expressing cells. Cells were routinely passaged and maintained in T75 flasks and were grown at 37 °C, 5% CO_2_, in a humidified incubator and tested for *Mycoplasma* once in every 2 months. Details of chemical inhibitors used in the study are as follows: PSI7977 (Sofosbuvir, HCV NS5B polymerase inhibitor, Adooq Biosciences, Irvine, CA, USA, A11529), KU-55933 (ATMi, TOCRIS, Bristol, UK, 3544), VE-821 (ATRi, Axon, Axon Medchem, Groningen, The Netherlands, 1893), SRA-737 (CHK1i, Selleckchem, Huston, TX, USA, S8253). Treatment conditions are mentioned in the text and figure legends.

### 2.2. Patient Samples

Archived diagnostic formalin-fixed paraffin-embedded (FFPE) liver biopsies from patients with HCV with or without associated HCC and non-HCV without HCC were obtained from the Newcastle Cancer Centre Biobank. All patients had provided written consent for the use of their tissues for research purposes. Ethical approval was obtained for the use of FFPE CHC patient biopsies (study reference: NAHPB-126) and non-HCV patient biopsies (biobank reference: 116370) by the National Research Ethics Service (NRES) Committee North East (REC ref: 12/NE/0395), sponsored by NUTH Trust R&D (Ref: 6579).

### 2.3. Plasmid Transfection

The indicated cells were seeded onto 6-well-plates and transfected the next day at approximately 80–90% confluence with appropriate plasmids using TransIT-LT1 transfection reagent (Mirus Bio., Madison, WI, USA, MIR 2304) in a 1:3 ratio as per the manufacturer’s instructions.

### 2.4. Retrotransposition Assay

Two different reporter systems were used to assess the active retrotransposition rates in the cell lines. The plasmids used for the assays were a kind gift from Dr Jose Luis Garcia-Perez, Institute of Genetics and Cancer, University of Edinburgh.

(a) EGFP as a reporter system: Cells were seeded onto a 6-well plate format and the following day were transfected with 1 µg of retrotransposition reporter plasmid containing an EGFP cassette interrupted by an intron so that EGFP expression occurred only after a successful cycle of retrotransposition, leading to EGFP splicing and integration into the genome. Either 99-GFP-LRE3 or 99-UB-GFP-LRE3 (retrotransposition competent) plasmid containing an EGFP-based retrotransposition cassette and puromycin resistance gene or corresponding negative control JM111 plasmid (retrotransposition incompetent due to a 260ARR-AAA262 mutation nearer to the C-terminus of ORF1) were used [28,29]. Five days after transfection, FACS analysis of live cells was carried out to check EGFP-positivity using an NxT Attune flow cytometer and with data analysis accomplished using FCS Express 7 software. SSC-FSC scatter plots were used to select single-cell populations and EGFP-positive populations were gated based on untransfected and JM111-transfected cells. In parallel, the cells were pelleted for genomic DNA extraction using a DNeasy Blood and Tissue kit (Qiagen GmbH, Hilden, Germany, 69504) that was further analysed quantitatively by Taqman qPCR using 2× Taqman genotyping qPCR master mix (Thermo Fisher Scientific Inc., Waltham, Massachusetts, USA, 4371353) and QuantStudio™ 7 flex Real-Time PCR system to determine the EGFP insertion rate. EGFP copy numbers were normalised to RNaseP and calculated using the *ΔΔCt* quantification method. The primers and probe set utilised were: GFP_F: GAAGAACGGCATCAAGGTGAAC; GFP_R: GGTGCTCAGGTAGTGGTTGTC; GFP Probe: (6FAM)-AGCGTGCAGCTCGCCGACCA(BHQ1) and RNaseP copy number assay with VIC-MGB probe (Applied Biosystems Inc., Waltham, MA, USA 4401631).

Where mentioned, retrotransposition was carried out in the presence of the indicated DDR pathway inhibitor in Huh7 cells. The inhibitors were added 3 days after transfection, along with 2 µg/mL puromycin to enrich for transfected cells. The treatment continued for a further 3–4 days. A no-inhibitor control with only puromycin treatment was always run in parallel.

(b) Blasticidin resistance as a reporter system: Huh7 and Huh7-J17 were seeded onto a 6-well-plate format and next day transfected with 1 µg wild-type L1 retrotransposition plasmid (pJJ101/L1.3), containing a blasticidin-based retrotransposition cassette or a mutant L1 retrotransposition plasmid (pJM105/L1.3mut, with a missense mutation (D702Y) in the reverse transcriptase domain of the ORF2 protein [30], or pcDNA6.1-blast plasmid as a positive control for the selection. Five days after transfection, the cells were harvested and transferred to 10-cm dishes with 4 µg/mL of Blasticidin (Sigma, SBR00022). Fresh blasticidin media was added every 3 days and the selection continued for a further 14–21 days to select blasticidin resistant colonies. To quantify L1 retrotransposition, the colonies were fixed with methanol and stained with 0.4% crystal violet (Sigma, C0775). Colonies were counted using an automated colony counter (Oxford Optronix, Milton, Abington, UK).

### 2.5. DNA Damage Repair Plasmid Re-Joining Assays

The homologous recombination repair (HRR) and non-homologous end joining repair (NHEJ) activity of the indicated cell lines was assessed by means of plasmid re-joining assays using pDRGFP or pimEJ5GFP reporter plasmids with pCBASce1, respectively, following established protocols in our lab [31]. In brief, pDRGFP contains an in vivo homologous recombination substrate that is composed of two differentially mutated GFP genes oriented as direct repeats and separated by a drug selection marker that can be excised by I-SceI. pCBASce1 expresses the I-SceI endonuclease that introduces a DSB at an I-SceI site. Upon successful repair, GFP is expressed in the cells and thus is an indicator of HRR. The pimEJ5GFP plasmid is an I-SceI-based chromosomal break reporter for NHEJ. In this reporter, end joining between two distal tandem I-SceI recognition sites restores an EGFP expression cassette, caused by deletion of the intervening pgkPURO cassette; thus, GFP expression is an indicator of NHEJ.

### 2.6. X-ray Irradiation Sensitivity Assay

Indicated cells were seeded onto 10-cm dishes and exposed to 2 Gy or 4 Gy X-ray the following day using Gulmay X-ray Generator (Model No. RS320, Serial No. GM0092, max 320 kV, Surrey, UK). Untreated cells were used as controls. The media was changed 24 h after X-ray exposure and then dishes were left undisturbed for 2–3 weeks to develop colonies. The colonies were fixed with methanol and stained with 0.4% crystal violet and measured using an automated colony counter. Plating efficiency % was calculated as
(no. of colonies formed/no. of cells plated) × 100(1)
and survival fraction % was calculated as
(plating efficiency in treatment group/plating efficiency in untreated conditions) × 100.(2)

### 2.7. Immunohistochemistry (IHC)

L1orf1p IHC was performed on a Ventana Discovery XT system, using standard protocol. In short, antigen retrieval was performed using Discovery CC1 buffer (Roche Diagnostics GmbH, Mannheim, Germany 06414575001 (950-500)), followed by incubation with the primary antibody against L1orf1p (1:2000, Mouse Monoclonal, MABC1152, Merck, Darmstadt, Germany), followed by anti-mouse-HRP secondary antibody (Roche 05266556001 (760-150)). An expert liver pathologist assessed the staining.

### 2.8. Western Blot Analysis

Western immunoblotting of whole-cell lysates was performed as described previously [32]. The primary antibodies used were anti-NS5A (mouse monoclonal, a kind gift from Charles M. Rice, Rockfeller University, New York , NY, USA), anti-L1orf1p (mouse monoclonal, Merck, MABC1152) and anti-GAPDH (rabbit monoclonal, Sigma, SAB2108266).

### 2.9. Luciferase Assay

The cells were lysed with passive lysis buffer (Promega, Madison, WI, USA) and luciferase activity was measured using luciferase assay reagent (Promega, E1500) and an Omega plate reader as per the instructions.

### 2.10. Bioinformatics

Human HCC RNAseq data were downloaded from The Cancer Genome Atlas Hepatocellular Carcinoma (TCGA-LIHC) project and RNAseq data of CHC patients and healthy controls GSE84346 were obtained from the NCBI GEO database. The reads were mapped to the human L1-Ta sequence (5′UTR-promoter, Genbank: L19092) by BLAT alignment using an in-house algorithm to obtain L1 counts (Python script can be provided upon request). The counts were normalised by the total number of reads in each library and expressed here as counts per million.

### 2.11. Statistical Analysis

GraphPad Prism software (GraphPad 8.0 and 9.0, San Diego, CA, USA) was used for statistical analysis. *p*-values < 0.05 were considered significant. Mean ± standard errors are shown in figures where applicable. * = *p* < 0.05, ** = *p* < 0.01, *** = *p* < 0.001, **** = *p* < 0.0001. Data were analysed using the one-sample *t*-test (for fold change), Student’s *t*-test (2 groups) or one- or two-way ANOVA with Tukey’s multiple comparison correction when required (3 groups).

## 3. Results

### 3.1. L1 Expression Is Upregulated in Non-Tumour Tissue of Patients with Chronic Hepatitis

To evaluate L1 activation in the liver of chronic hepatitis C patients prior to the development of HCC, we analysed a publicly available RNAseq dataset for a cohort of CHC patients (*n* = 14) and control (healthy, *n* = 6) individuals (GSE84346). There was significant upregulation of L1 transcripts in the livers of CHC patients (Figure 1A). Likewise, interrogation of the RNAseq dataset of the TCGA-LIHC study revealed the upregulation of L1 expression in the non-tumour livers of patients with a history of viral hepatitis compared to patients with no history of any known HCC risk factors (Figure 1B). However, upon cancer development, L1 was found to be upregulated in all the HCC cases irrespective of the underlying aetiology (Figure 1B). We also evaluated the presence of L1orf1-encoded protein (L1orf1p) expression by IHC in HCV-infected liver biopsies from our own biobank (Ref. no. NAHPB-126). Again, L1orf1p expression was observed in the non-tumour tissue of some individuals (six out of 11, ~54%), years before HCC development. In two of the cases with subsequent diagnostic HCC biopsy tissue available, the earlier non-tumour L1 status matched that of the HCC (i.e., positive remained positive and negative remained negative) (Figure 1C). We also analysed the pre-HCC liver biopsies in the case of non-viral fatty liver disease patients with alcohol related liver disease and metabolic syndrome from our own biobank (reference: 116370). One out of five (20%) non-tumour livers exhibited L1orf1p positivity. Similarly, there is a trend of an increase in L1 transcripts in the non-tumour tissue of patients with alcohol-related HCC, but it did not reach statistical significance when compared to individuals with no history of any known HCC risk factors (TCGA-LIHC RNAseq data, Appendix A). Hence, the data demonstrate that L1s can be activated in a chronically diseased pre-neoplastic liver, especially when associated with HCV infection, and may contribute to cancer development.

### 3.2. HCV Activates L1 Retrotransposition

To evaluate the direct influence of HCV infection on L1 retrotransposition, Huh7-J17 cells stably expressing an HCV sub-genomic replicon (HCV genome without envelope proteins fused with the puromycin-resistance cassette and luciferase expression cassette) were employed [26,27]. Huh7-J17 cells were also positive for luciferase activity, which decreased upon treatment with PSI7977 (sofosbuvir, a NS5B polymerase inhibitor that inhibits virus replication) in a dose-dependent manner (Figure 2A), confirming the presence of the HCV replicon. The presence of the viral replicon was also further confirmed by Western blotting, showing the expression of the HCV NS5A protein (Figure 2B top panel and Appendix A). Similarly to CHC patient livers, upregulation of L1orf1p was observed in Huh7-J17 compared to Huh7 cells (Figure 2B bottom panel and Appendix A).

Next, we compared the retrotransposition efficiency of the Huh7-J17 cell line with the corresponding parental or naïve Huh7 cells via an in vitro EGFP-based retrotransposition assay assessed using FACS and PCR analysis (Figure 2C). A significant increase in cells undergoing active retrotransposition was observed in Huh7-J17 cells compared to the Huh7 control cells, as indicated by an increased number of EGFP-positive cells by FACS analysis (fold change ~3.2, Figure 2D and Appendix A). The insertion of the L1-EGFP plasmid in the genome was further verified via genomic PCR to detect the intron-less genomic GFP sequence and was quantified via Taqman qPCR, revealing a ~2.9-fold increase in retrotransposition efficiency in Huh7-J17 cells with the HCV replicon compared to the naïve cells (Figure 2E and Appendix A). The influence of HCV replication on active retrotransposition was further independently verified using a blasticidin-based retrotransposition assay, wherein the attainment of blasticidin resistance acts as a marker of active retrotransposition [29]. Again, a significant increase in blasticidin-resistant colonies were observed in Huh7-J17 compared to Huh7, indicating an increased retrotransposition rate in the cells in the presence of HCV RNA (fold change: 6.8 ± 1.36, *n* = 3, Figure 2F,G). As expected, cells transfected with the L1.3-mutant plasmid exhibited no colonies, proving that the colonies obtained in L1.3wt set were retrotransposition-specific. Moreover, cells transfected with the pcDNA6.1-blast control plasmid developed a complete lawn in both the cell lines due to the survival of all the transfected cells (Figure 2F). However, when seeded at a limited density, the ability to attach and survive under blasticidin selection of Huh7-J17 cells transfected with the control pcDNA6.1 blast plasmid was about 4.5-fold lower than Huh7 cells under similar conditions (Appendix A). In spite of lower colonies in Huh7-J17 cells transfected with pcDNA6.1, there were higher colonies in Huh7-J17 cells transfected with L1.3wt compared to Huh7 cells in respective conditions, thus supporting the conclusion that the active L1 retrotransposition rate is higher in Huh7-J17 cells compared to Huh7 cells.

### 3.3. HCV Potentially Upregulates L1 Retrotransposition via the Inhibition of DNA Damage Repair Pathways

For a successful round of L1 retrotransposition, an L1 element has to transcribe and translate an mRNA to obtain its two encoded proteins (L1orf1p and L1orf2p), which then bind with the transcript, forming a L1 ribonucleoprotein complex (RNP) [33]. The L1-RNP complex in the nucleus reverse transcribes the transcript and inserts a new genomic location mainly via a molecular mechanism known as target-primed reverse transcription (TPRT) [34,35]. In short, the L1orf2p-encoded endonuclease domain makes a DNA nick on one strand of the host genome at the AT-rich consensus sequence (5′-TTTT/AA-3′ consensus) and the resulting 3′-OH end is extended by the reverse transcriptase domain of L1orf2p that makes a DNA copy from the L1 RNA, used as a template. Less frequently, L1 integration can also occur independently of L1-endonuclease activity, using pre-existing chromosomal DNA breaks [36]. Hence, the process of retrotransposition is controlled at various steps by several nuclear and cytoplasmic proteins [37]. There are several reports demonstrating the effect of DNA repair factors on L1 retrotransposition but observations are inconsistent between studies; for example, an activating role as well as an inhibitory role of ATM on L1 retrotransposition has been reported [38,39]. Recently, a systematic search for host factors affecting L1 mobility was carried out via whole genome siRNA screening and identified double-strand DNA break (DSB) repair, especially BRCA1-dependent homologous recombination repair (HRR), and Fanconi anemia (FA) factors as potent inhibitors of L1 activity in HeLa cells [40].

Since HCV is known to impair DNA repair mechanisms, we used a colonogenic assay to first compare the sensitivity of Huh7 and Huh7-J17 cells in response to a potent DNA damaging agent, i.e., ionising radiation (IR), which causes a complex spectrum of both DNA single-strand breaks (SSBs), as well as DSBs. Huh7-J17 cells were found to be significantly more sensitive to IR compared to parental Huh7 cells (Figure 3A,B). It is noteworthy that the non-irradiated colonogenic efficiency of Huh7-J17 cells was about half that of the parental Huh7 cells (~26% Huh7-J17 versus ~56% for Huh7), indicating an intrinsic stress in these cells due to the presence of HCV (Figure 3C).

Compared to SSBs, DSBs are more lethal to cells if unrepaired. There may be roles for either HRR or non-homologous end joining (NHEJ) pathways in the repair of IR-induced DSBs in these cells. Thus, we next used plasmid-based reporter assays to test HRR and NHEJ pathways in Huh7 and Huh7-J17 cells. The reporter assays revealed significant downregulation of the HRR pathway in Huh7-J17 cells compared to Huh7 parental cells (Figure 5E). Of note, there was no significant influence of the HCV replicon observed on the NHEJ pathway (Figure 5F). Hence, we postulated that the HRR pathway may influence the regulation of L1 retrotransposition in Huh7 cells.

We therefore examined the effect of blocking the DNA damage response enzymes on L1 retrotransposition frequencies in the Huh7 cell line using small molecule inhibitors that target ATM (KU-55933, 10 µM), ATR (VE-821, 1 µM) and CHK1 (SRA-737, 1 µM). A significant increase in L1 retrotransposition efficiency upon inhibition of ATR and CHK1 was observed in Huh7 cells (Figure 4A,B; effectiveness of the inhibitors against their respective targets is shown in Appendix AA,B). However, there was no influence on the L1orf1p level in Huh7 cells upon treatment with CHK1i (Appendix AC). ATR and CHK1 are known to play an important role in the maintenance of DNA integrity in the face of DNA damaging insults, principally through their involvement in the HRR response as well as cell cycle checkpoints [41]. This supports the hypothesis that the HRR pathway is the main DDR pathway involved in the regulation of L1-mediated genomic instability.

### 3.4. Influence on L1 Retrotransposition and Its Consequences Persists after Viral Clearance

A number of genes are reported to be dysregulated by HCV via epigenetic mechanisms continuing beyond viral clearance [18,19]. Likewise, the consequences of L1-mediated somatic mutagenesis will continue in cells beyond viral clearance and may give rise to new lineages of active L1 copies. To address whether active viral infection is essential to the upregulated retrotransposition rate we observed in vitro, we treated Huh7-J17 cells with PSI7977 (sofosbuvir, a NS5B polymerase inhibitor that inhibits virus replication). As shown in Figure 2A, PSI7977 treatment produced a dose-dependent decrease in HCV replicon levels in Huh7-J17 cells, as judged by the overall luciferase activity of the cells. The 10 µM dose was selected and Huh7-J17 cells were treated with 10 µM of PSI7977 in the absence of puromycin for 3 weeks, to mimic DAA treatment in order to generate a cell line that is clear of the virus, to be used as a post-HCV clearance model (Huh7-J17+PSI7977). The loss of the HCV replicon from the cells was confirmed by checking the puromycin sensitivity of the cells post PSI7977 treatment. As expected, 100% cell death was observed in Huh7-J17+PSI7977 cells upon puromycin treatment (2 µg/mL for 48 h). In addition, no NS5A protein was detected in the Huh7-J17+PSI7977 whole cell lysate, confirming the clearance of the HCV replicon from the cells (Figure 5A). Moreover, the L1orf1p level dropped back to the parental Huh7 level upon viral clearance in Huh7-J17+PSI7977 cells (Figure 5A and Appendix A), indicating that the presence of the HCV transcript or its encoded protein(s) is essential to upregulate L1orf1p.

Next, EGFP- and blasticidin-based retrotransposition assays were carried out in Huh7-J17+PSI7977 cells and compared with Huh7 and Huh7-J17 cells. Although lower than Huh7-J17, the level of active retrotransposition remained upregulated in Huh7-J17+PSI7977 cells compared to Huh7 cells (fold change 2.48 ± 0.19, *n* = 3, *p* < 0.05 one-sample *t*-test Figure 5B,C and Appendix A), thus indicating that the influence of HCV on L1 retrotransposition involves a ‘hit-and-run’ mechanism via pathway(s) which remain dysregulated in cells even after viral clearance. The increased retrotransposition rate in Huh7-J17+PSI7977 cells compared to Huh7 cells is further supported by the fact that the actual ability of the cells to attach and survive under blasticidin selection conditions when transfected with the pcDNA6.1-blast control plasmid and seeded at a limited density was about 2-fold lower compared to Huh7 cells (Appendix A).

The radiation sensitivity of Huh7-J17+PSI7977 cells was restored to the parental Huh7 cell level (Figure 5D and Appendix A), with the plating efficiency also comparable to parental Huh7 cells (49% ± 5% for Huh7-J17+PSI7977 cells), indicating that the stress induced by HCV replication and the dysregulation of DNA damage repair pathways returned to a steady state level upon viral clearance. Plasmid-based DNA damage response assays confirmed the restoration of the HRR pathway in Huh7-J17+PSI7977 cells (Figure 5E). Similar to Huh7-J17 cells, there was no significant difference in NHEJ pathway between Huh7-J17+PSI7977 and Huh7 parental cells (Figure 5F). Hence, the underlying mechanisms regulating active retrotransposition were different in the presence of active HCV infection versus post-viral clearance, warranting further investigation.

## 4. Discussion

Our study provides evidence of L1 activation in the chronically infected HCV liver and demonstrates that HCV infection can influence the L1 retrotransposition process. In addition, we have demonstrated that the rate of retrotransposition remains enhanced even after viral clearance, when compared to cells with no viral exposure. We have also demonstrated that the presence of HCV exerted stress on the cells, rendering them more sensitive to DNA damage. This is in line with a previous report, in which HCV-encoded NS5a is shown to bind to RAD51AP1, leading to the impairment of the RAD51/RAD51AP1/UAF1 trimeric complex, thus impairing DNA repair and resulting in increased hypersensitivity to DNA damaging agents [14]. On the other hand, HR factors are known to restrict retrotransposition, potentially by sterically blocking the formation of the second DNA nick during L1 retrotransposition, perhaps by recruiting ssDNA-coating proteins RPA1 and Rad51 [40]. In summary, we observed increased retrotransposition in the HCV-infected cells, with impaired DDR pathways as the potential mechanisms. However, how the influence of HCV infection on L1 retrotransposition continues even after viral clearance is not known.

Of note, Schobel et al. recently reported a negative (rather than positive) influence of HCV on the retrotransposition process [42], observing a reduction in L1 retrotransposition in both Huh7.5 cells, which contains a mutation (Thr-55-Iso) in the RIG-I (*DDX58*) gene and in Huh7 cells in the presence of HCV infection, which was attributed to an accumulation of L1orf1p in the stress granules in the presence of the virus in their experiments. More in keeping with our own studies, they observed an increase in retrotransposition on overexpression of the HCV core protein, with upregulation of L1orf1p levels in the cells in the presence of HCV. Discrepancies in the rates of retrotransposition are likely due to differences in the model systems utilised, with both studies implicating L1s as a source of genomic instability and recognising that cells have evolved multiple mechanisms to suppress these elements and block retrotransposition [37]. HCV differentially modulates several L1 inhibitory pathways, including the activation of autophagy and interferon response pathways, whereas it impairs DNA damage repair pathways, especially HRR. Hence, there is a delicate balance between these various pathways, which determines the success or failure of active retrotransposition in a cell. The viral load can be a major determinant of the extent of dysregulation of these processes and the final outcome of the altered rate of active retrotransposition. In addition to HCV, the influence of other viruses on L1—such as Kaposi’s sarcoma (KS)-associated herpesvirus—leading to the upregulation of retrotransposition has been demonstrated (KSHV) [43], with contradictory reports about the influence of human immunodeficiency virus (HIV) [44,45,46].

The present work extends the previous observation by Shukla et al. [21] that demonstrated L1 activation and active retrotransposition in HCV-associated HCC by indicating the possibility of L1 activation and HCV-supported active retrotransposition in the preneoplastic liver. This suggests another potential contributing factor to be the underlying cause of hepatocarcinogenesis in CHC patients as L1-mediated potential driver mutations have been implicated in various cancer types [22]. Before the DAA era, CHC patients were treated with IFN therapy and type 1 interferons restrict L1 retrotransposition [47]. However, the influence of DAAs on L1 activity is not known and is worth investigating further. In addition, the comparison of the frequency of L1-mediated genomic rearrangements in HCV-HCC developed in IFN therapy patients versus DAA therapy patients can shed light further on the role of L1 in HCC development in CHC patients. In addition, L1 retrotransposition can be inhibited by means of anti-retroviral drugs [48,49]. Many of these are currently in clinical use, such as nucleoside reverse transcriptase inhibitors (NRTIs), e.g., lamivudine for HBV and HIV therapy and non-nucleoside reverse transcriptase inhibitors (NNRTIs), e.g., efavirenz for HIV treatment [50]. Hence, it will be worth interrogating the effect of these drugs on the rate of HCC development in patients co-infected with HBV-HCV (1–15% of world population [51]) and HIV-HCV (~6.2% HIV-infected individuals [52]) and evaluate if the HCC that developed in these patients is any different from only HCV-associated HCC, especially in terms of L1-mediated genomic rearrangements.

Globally an estimated 130–170 million people (2–3% of the world’s population) are living with HCV infection and more than 350,000 die of HCV-related conditions (including HCC) per year [53]. Even if the sustained viral response (SVR) is attained via DAA therapy, the risk of cancer persists in patients with advanced fibrosis or cirrhosis and these individuals require ongoing surveillance for HCC [5,54,55]. Under the current guidelines, monitoring for HCC with liver imaging and blood AFP (alpha-fetoprotein) biomarker levels should be performed twice a year indefinitely post-SVR in patients with HCV cirrhosis [56]. Hence, understanding the risk of progression in an individual patient would have enormous value to help develop stratified approaches to surveillance, better screening tools and prevention. Progress in this field has been relatively slow, although genetic studies show promise in the polymorphisms of some genes, such as programmed death receptor 1 (*PDCD1*) [57], patatin-like phospholipase domain-containing 3 (*PNPLA3*) [58] and transmembrane-6 superfamily member-2 (*TM6SF2*) [59], in combination with other risks (sex, age, BMI, alcohol, type 2 diabetes), may ultimately play a role in stratified surveillance for chronic liver disease patients [60]. For CHC patients, L1 activation should be explored further as an HCC risk factor. Furthermore, as L1 activation can be monitored by measuring the methylation status of L1 promoters in circulating cell-free DNA isolated from peripheral blood [61,62], simple tests may be sufficient, without the need for the analysis of tissue biopsies. Overall, the present study identifies L1 as a novel potential biomarker for HCC risk prediction and a potential novel target to impede hepatocarcinogenesis in CHC patients.

## 5. Conclusions

We have demonstrated that L1 expression in liver tissue can be upregulated in chronic liver disease patients, years before HCC development, and this is especially observed in CHC patients. L1 activation could be due to epigenetic changes as a field effect in the liver due to chronic inflammation and increased oxidative stress, associated with chronic liver disease. We have also demonstrated that HCV impairs the HRR pathways and activates L1 retrotransposition. HCV also upregulated L1orf1p expression. Treatment of Huh7 cells, an HCC cell line, with Chk1 inhibitor increased the L1 retrotransposition rate but did not influence the L1orf1p level; hence, we speculate its mechanism to occur via DDR-independent pathways. The exact causative relationship between these factors is not known and warrants further investigation.

## Figures and Tables

**Figure 1 cancers-13-05079-f001:**
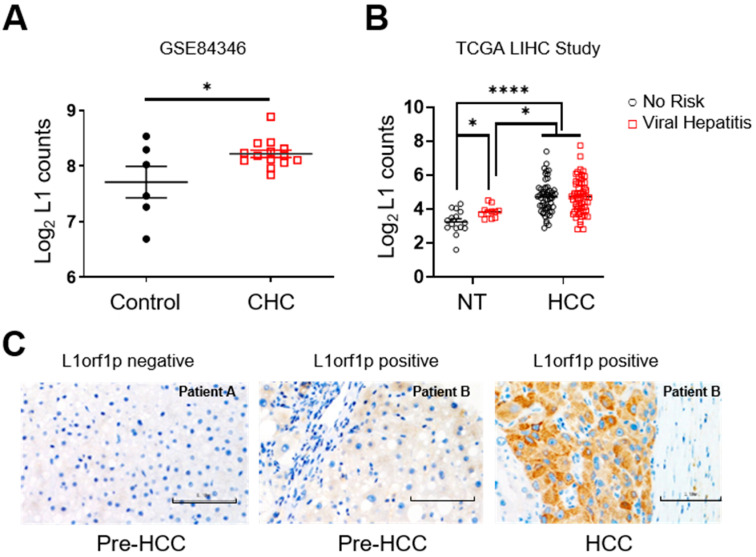
HCV activates L1 expression in non-tumour livers. Graphs represent normalised L1 transcript counts in the liver of chronically infected HCV patients versus the healthy volunteer control group (**A**), in the non-tumour liver and HCC tissue of HCC patients with indicated aetiologies; no risk = no history of any known HCC risk factors (**B**). Representative L1orf1p IHC images of CHC patient liver biopsies pre-HCC and HCC, where blue represents hematoxylin stained nuclei and brown staining is representative of presence of L1orf1p. Scale bar represents 0.1 mm (**C**). * *p* < 0.05, **** *p* < 0.0001, unpaired *t*-test (**A**) and two-way ANOVA (**B**). Key: CHC = Chronic Hepatitis C patients; NT = Non Tumour; TCGA = The Cancer Genome Atlas; LIHC = Liver Hepatocellular Carcinoma; HCC = Hepatocellular Carcinoma; HCV = Hepatitis C Virus; L1 = Long interspersed repeat element 1; L1orf1p = L1 open reading frame 1 encoded protein.

**Figure 2 cancers-13-05079-f002:**
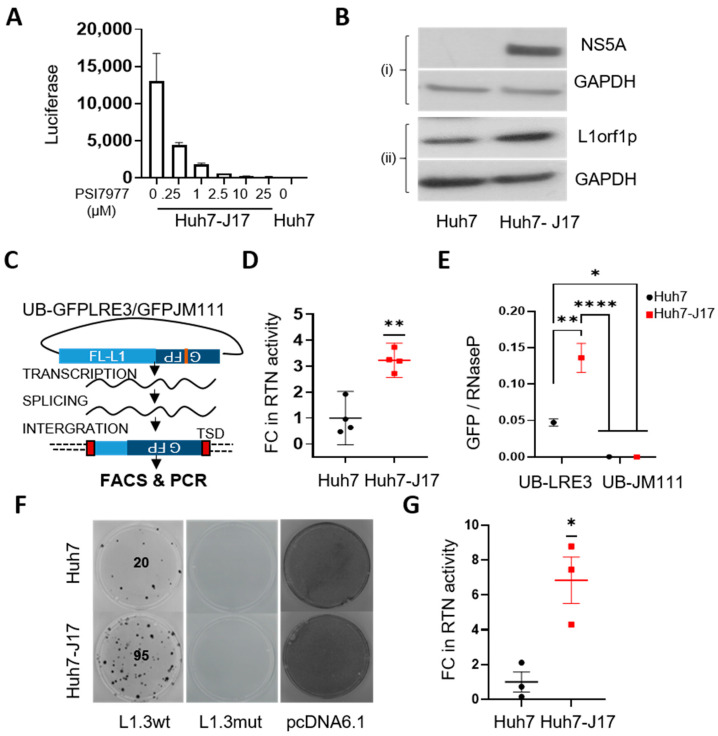
HCV activates L1s. Luciferase expression levels (indicator of HCV replicon level) in Huh7-J17 cells upon treatment with indicated doses of PSI7977 (sofosbuvir) for 48 h. Parental Huh7 lysate was used as a negative control. Bars represent the mean ± SE of 2 technical repeats (**A**). Western blot images of whole-cell lysates of Huh7 and Huh7-J17 cells probed with the indicated antibodies. GAPDH was used as a loading control (**B**). Schematic representation of the retrotransposition assay with EGFP indicator cassette. UB denotes the presence of the UB promoter before the L1-retortransposition cassette. The JM111 plasmid contains a mutation in L1ORF1 (RR260-261AA) of the LRE3 sequence, rendering it retrotransposition-incompetent and it is thus used as a negative control (**C**). The results of the retrotransposition assay in Huh7 cells in the presence and absence of HCV: graph representing fold change (FC) in the number of retrotransposition (RTN)-positive cells observed via FACS analysis 5 days after transfection with UB-GFPLRE3 plasmid, where the negative gate was set using cells transfected with UB-GFPJM111 plasmid, *n* = 4 independent repeats (**D**), quantification of GFP insertions in the genome using RNaseP as a control, *n* = 3 technical replicates of one representative Taqman qPCR assay (**E**) and blasticidin-resistant colonies representative of active retrotransposition events, visualised via crystal violet staining 3 weeks after selecting cells with blasticidin. Blasticidin selection was started 5 days after transfection with a retrotransposition plasmid-containing blasiticidin indicator cassette (L1.3wt or negative control plasmid L1.3mut, which contains a missense mutation (D702Y) in the reverse transcriptase domain of L1ORF2-protein, rendering it retrotransposition incompetent, and pcDNA6.1, constitutively expressing a blasticidin resistance cassette as a positive control for transfection efficiency and blasticidin resistance). The image is representative of 3 independent repeats done in duplicates. Numbers represent the number of colonies in the plate (**F**) and the graph represents the fold change in number of colonies representing active retrotransposition taking Huh7 cells as control (**G**). * *p* < 0.05, ** *p* < 0.01, **** *p* < 0.0001, one-sample *t*-test (**D**,**G**) and one-way ANOVA with Tukey’s multiple testing correction (**E**). Detailed information about the Western blotting can be found in Appendix A. Key: GFP = Green Fluorescent Protein; RTN = Retrotransposition; FC = Fold Change; TSD = Target Site Duplication; FACS = Fluorescence Activated Cell Sorting; FL-L1 = Full Length L1 element; NS5A = nonstructural protein 5A; L1orf1p = L1 open reading frame encoded protein; GAPDH = Glyceraldehyde 3-phosphate dehydrogenase; UB = Ubiquitin; HCV = Hepatitis C Virus; SE = Standard Error.

**Figure 3 cancers-13-05079-f003:**
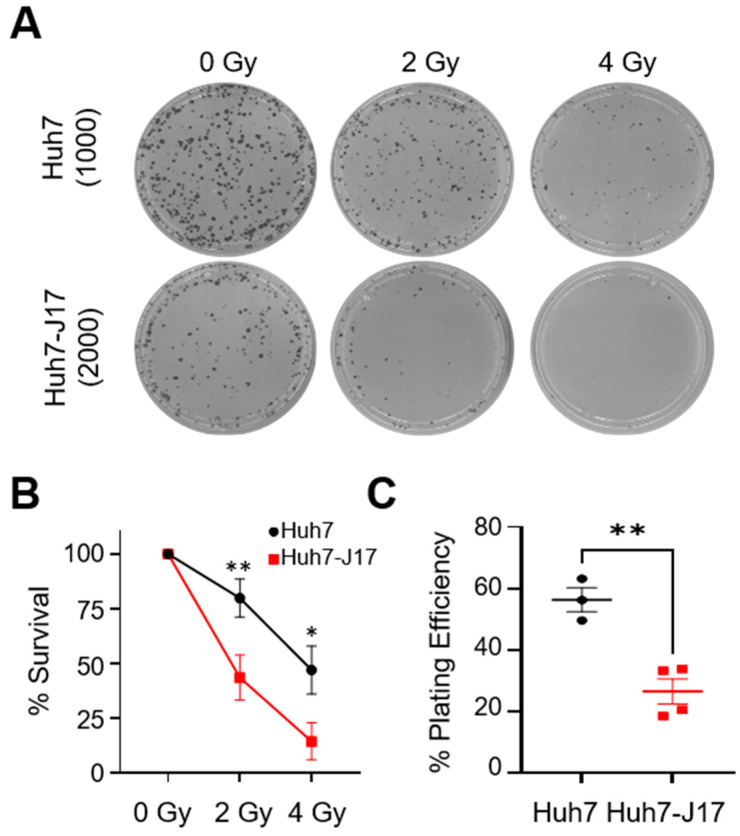
HCV impairs DNA damage repair pathways in Huh7 cells. Plates representing colonies formed without any treatment (0 Gy) and after cells were exposed to 2 Gy or 4 Gy X-ray radiation, as visualised by crystal violet stain. Image is representative of 3 independent repeats done in duplicates. Numbers represent initial cells seeded (**A**). % clonogenic cell survival plots (**B**) and plating efficiency % (**C**) of Huh7 and Huh7-J17 cells, indicating the increased sensitivity of cells to irradiation in the presence of Hepatitis C Virus (HCV). * *p* < 0.05, ** *p* < 0.01. 2-way ANOVA with Tukey’s multiple test correction (**B**) and unpaired *t*-test (**C**).

**Figure 4 cancers-13-05079-f004:**
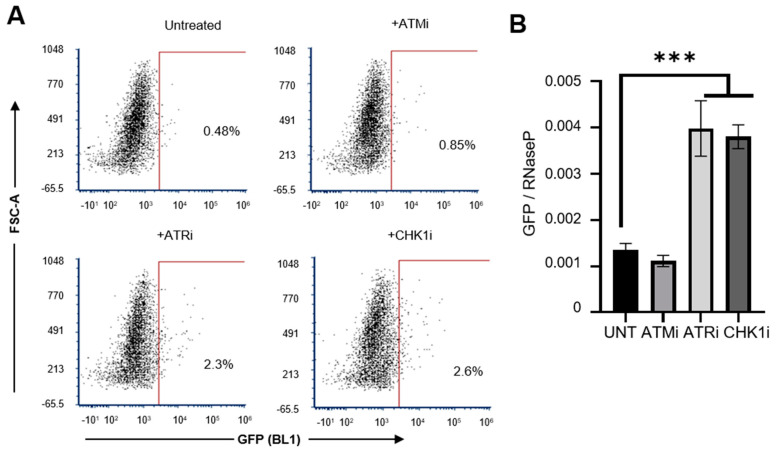
Regulation of active retrotransposition in Huh7 cells by DDR response pathways. Level of L1 retrotransposition was measured using FACS and Taqman qPCR assays, following the transfection of Huh7 WT cells with the GFP-LRE3 plasmid, at the basal level or upon treatment with the indicated DDR pathway inhibitors. Numbers on the dotplots represent % EGFP positive cells (representative of cells undergoing active retrotransposition). The positive gate is marked by red lines. (**A**). Quantification of GFP insertions in the genome using RNaseP as a control, *n* = 3 technical replicates (**B**). *** *p* < 0.001, one-way ANOVA with multiple comparisons. Key: FSC-A = Forward Scatter Area; DDR = DNA Damage Response; qPCR = Quantitative Polymerase Chain Reaction; ATMi = Ataxia Telangiectasia Mutated kinase inhibitor; ATRi = Ataxia-Telangiectasia and Rad3-related kinase inhibitor; CHK1i = Checkpoint kinase 1 inhibitor; UNT = Untreated.

**Figure 5 cancers-13-05079-f005:**
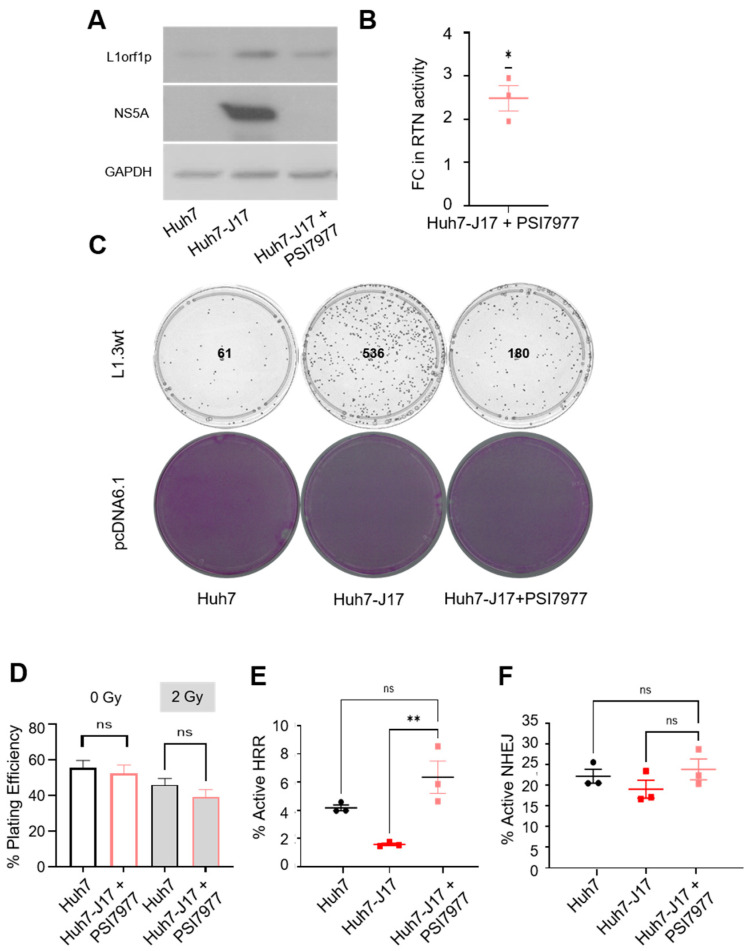
Influence of HCV on L1 retrotransposition continues even after viral clearance. Western blot analysis on whole cell lysates of indicated cells to confirm the clearance of Hepatits C Virus (HCV) replicon and the L1orf1p level 3 weeks after treatment with 10 µM PSI7977. Tubulin was used as a loading control (**A**). Graph represents the fold change (FC) in the number of colonies representing active retrotransposition (RTN) in Huh7-J17+PSI7977 cells compared to Huh7 cells (**B**). Plates showing retrotransposition rates in indicated cell lines as assessed via blasticidin-based selection of resistant colonies. Image is representative of 3 independent repeats. Numbers represent number of colonies in the plate (**C**). Plating efficiency (%) of the indicated cells when exposed to 2 Gy X-rays or not (**D**). Graphs showing the percentage of GFP-positive cells representing active HRR (**E**) and NHEJ (**F**) repair in the indicated cell lines. ns = non-significant (*p* > 0.05), * *p* < 0.05, ** *p* < 0.01 by one-sample *t*-test (**B**), unpaired *t*-test (**D**) and one-way ANOVA (**E**,**F**). Detailed information about the Western blotting can be found in Appendix A. Key: L1orf1p = L1 open reading frame encoded protein; NS5A = nonstructural protein 5A; GAPDH = Glyceraldehyde 3-phosphate dehydrogenase.

## Data Availability

The data presented in this study are openly available in the NCBI GEO repository (https://www.ncbi.nlm.nih.gov/geo/query/acc.cgi?acc=GSE84346, accessed on 8 October 2019) and can be obtained by permission from the GDC Portal (https://portal.gdc.cancer.gov/projects/TCGA-LIHC, accessed on 14 February 2018). The script used for the analysis can be requested from the corresponding author. All other data are contained with the article and Appendix A.

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
