# Peer review of "HCV Activates Somatic L1 Retrotransposition—A Potential Hepatocarcinogenesis Pathway"

_cancers, 2021, doi:10.3390/cancers13205079_

Round 1

Reviewer 1 Report

The authors have incorporated my suggestions and I find that the manuscript is improved. I think that this paper is now worth publishing in Cancers. 

Author Response

We would like to thank the reviewer for recommending the manuscript to be published in Cancers.

Reviewer 2 Report

I appreciate the efforts of the authors, the new version of the manuscript has improved. However, I still consider that the increase on L1 retrotranposition should be better demonstrated as it is the main evidence to support their conclusions. In fact, the control that they have included for the blasticidin-based retrotransposition assay is not conclusive (transfection with pcDNA6.1). It is not possible to distinguish colonies and, consequently, it is impossible to compare the ability to attach and form colonies from the different cell lines. Please, replace controls (Fig. 2F and 5C) with other wells where the colonies are countable and therefore useful as a normalizer to determine the retrotranposition rate. (see Fig. 3 from Moldovan JB, Moran JV. The Zinc-Finger Antiviral Protein ZAP Inhibits LINE and Alu Retrotransposition. PLoS Genet. 2015).

Author Response

We would like to thank the reviewer for appreciating the changes we have made to the manuscript so far. We understand the concern raised by the reviewer regarding differences between the cell lines in terms of their clonogenic abilities when blasticidin-based retrotransposition assay was performed. This was partly addressed in the manuscript showing reduced plating efficiency of Huh7-J17 cells compared to Huh7 cells (Fig 3A) while plating efficiency of Huh7-J17+PSI7977 was found to be comparable to Huh7 cells (Fig 5D). However, to address the issue fully as suggested by the reviewer, we have carried out transfection of all the 3 cell lines with pcDNA6.1-blast plasmid and evaluated their survival and plating efficiency. In short, 300,000 cells/well were seeded in a 6 well plate for each cell line and were transfected with 1µg pcDNA6.1-blast plasmid next day (as done for all the retrotransposition assays), one well per cell line was left as none-transfected control. 4 days after transfection, the transfected cells were harvested by trypsinisation and counted to seed 8000, 4000 and 1000 cells per 10cm dish for each cell line, the plating was done as 3 technical replicates with independent counting. The cells were selected with media containing blasticidin (4µg/ml) for the next 2 weeks alongside none-transfected control cells. The plates were then fixed and stained with crystal violet. As shown in Supplementary figure S5 in the revised manuscript, Huh7 cells exhibited the highest plating and survival efficiency (~9.7%; 919/8000, 375/4000, 82/1000) under blasticidin selection followed by Huh7-J17+PSI7977 (~4.6%; 449/8000, 145/4000, 47/1000) and then Huh7-J17 (~2.2%; 204/8000, 68/4000, 24/1000). No colonies were obtained in any of the none-transfected controls. Despite Huh7-J17 cells having the lowest plating and survival ability following control pcDNA6.1-blast transfection, the number of colonies obtained with the L1.3-blast retrotransposition (RTN) assay plasmid was highest in Huh7-J17 cells, confirming the upregulation of RTN rate in the Huh7-J17 cells. The data and its interpretation is added to the manuscript (see lines 292-302 and 414-418).

The reviewer has asked to replace the controls of Fig 2F and 5C but we do not agree to do so, as in the figures the control plates were run alongside the experiment and the same number of cells (~300,000) were seeded in the control plates and RTN plates.  Instead we have added the additional control experiment as a supplementary figure (Fig S5) and hope that this will clarify any doubts the reviewer has regarding our main conclusion that L1 retrotransposition is upregulated in Huh7 cells containing the HCV replicon.

Round 2

Reviewer 2 Report

The authors have addressed my main point carefully. I have no further comments at this time and approve publication.

This manuscript is a resubmission of an earlier submission. The following is a list of the peer review reports and author responses from that submission.

Round 1

Reviewer 1 Report

The manuscript by Sudhindar et al. shows that L1 retrotransposons are upregulated in chronic hepatitis C virus (HCV) hepatitis (CHC) patients and HCV potentially activates them by inhibition of DNA damage repair (DDR) pathways, especially homologous recombination repair (HRR) in double-strand DNA break (DSB) repair. Moreover, the manuscript shows that L1 retrotransposition remained upregulated in Huh7 cells even after viral clearance. The topic is interesting, the methods are adequate and up-to-date, and the paper is exceptionally well written. I recommend several revisions.

1) It is better to explain the principle of HRR and NHEJ reporters briefly in the materials and methods section.

2) Figure 2, panel F. Although the authors wrote that retorotransposition was increased by 6.8-fold, it is better to provide a graph depicting the level of L1 retrotransposition with the p-value.

3) Figure 5. Similar to the comment #2, please provide a graph of L1 retrotransposition with the p-value. I noticed that the authors conducted this experiment only twice, but more trials are required for statistical evaluation. It will help to understand how the underlying mechanisms were different (line 377).

4) Figure 5. What is the expression level of L1orf1p after viral clearance? Is it still upregulated or not?

5) Line 389, what “directly” means? Because L1 retrotransposition was affected even after viral clearance, it seems an “indirect” effect of HCV.

6) Supplementary Figure S6. This figure should be added to Figure 5 to better understand the restoration of the radiation sensitivity and the HRR pathway in Huh7-J17+PSI7977. Please check the data of Huh7 and Huh7-J17 in Figure S6B and C because the dots of Huh7 and Huh7-J17 are identical to those in Fig .3D and E.

7) In this study, the authors demonstrated that HCV impairs the HRR pathway and activates L1 retrotrasposition. Also, they found that HCV upregulates L1expression. However, the causative relationship of these three factors is not demonstrated at all. I recommend to conduct additional experiments that support the authors’ hypothesis. For example, orf1p expression after viral clearance and ATRi or CHKi treatment to Huh7-J17 cells.

8) Please provide the information of the mechanisms how HCV impairs the HRR pathway and how the HRR pathway regulates L1 retrotransposition and/or discuss the possibilities.

Author Response

The manuscript by Sudhindar et al. shows that L1 retrotransposons are upregulated in chronic hepatitis C virus (HCV) hepatitis (CHC) patients and HCV potentially activates them by inhibition of DNA damage repair (DDR) pathways, especially homologous recombination repair (HRR) in double-strand DNA break (DSB) repair. Moreover, the manuscript shows that L1 retrotransposition remained upregulated in Huh7 cells even after viral clearance. The topic is interesting, the methods are adequate and up-to-date, and the paper is exceptionally well written. I recommend several revisions.

We would like to thank the reviewer for finding the study interesting and recommending the changes to improve it. Please see our point-by-point response below.

1) It is better to explain the principle of HRR and NHEJ reporters briefly in the materials and methods section.

We have included the principle of the assay in the methods section, see lines 185-193.

2) Figure 2, panel F. Although the authors wrote that retrotransposition was increased by 6.8-fold, it is better to provide a graph depicting the level of L1 retrotransposition with the p-value.

As suggested by the reviewer, the data is now represented as a graph in figure 2 (2G).

3) Figure 5. Similar to the comment #2, please provide a graph of L1 retrotransposition with the p-value. I noticed that the authors conducted this experiment only twice, but more trials are required for statistical evaluation. It will help to understand how the underlying mechanisms were different (line 377).

 We have included one more repeat of the assay and updated the data in the text (see lines 402-403) and have provided a graph for this in figure 5 (5B).

4) Figure 5. What is the expression level of L1orf1p after viral clearance? Is it still upregulated or not?

Thank you for asking this pertinent question. The L1orf1p level is restored upon viral clearance. A Western blot image is included in figure 5 (5A).

5) Line 389, what “directly” means? Because L1 retrotransposition was affected even after viral clearance, it seems an “indirect” effect of HCV.

What we meant by ‘’directly’’ was influence of HCV in hepatocytes rather than an indirect influence of the liver microenvironment. However, we understand the concern raised by the reviewer and the misunderstanding this may create. Hence, have removed the word ‘directly’ from the sentence (this is line 427 in the revised version).

6) Supplementary Figure S6. This figure should be added to Figure 5 to better understand the restoration of the radiation sensitivity and the HRR pathway in Huh7-J17+PSI7977. Please check the data of Huh7 and Huh7-J17 in Figure S6B and C because the dots of Huh7 and Huh7-J17 are identical to those in Fig .3D and E.

As suggested by the reviewer Fig S6 is now included in Fig 5 (5E and 5F). Further we would like to clarify that the data for Huh7 and Huh-J17 was actually same in Fig 3D-E and S6, in S6 additional data of Huh7-J17+PSI7977 was included. Since now that the graphs (previous S6) are included in main figure we have deleted Fig 3D and 3E to avoid unnecessary repetition. Reference to the figures are updated accordingly.

7) In this study, the authors demonstrated that HCV impairs the HRR pathway and activates L1 retrotransposition. Also, they found that HCV upregulates L1expression. However, the causative relationship of these three factors is not demonstrated at all. I recommend to conduct additional experiments that support the authors’ hypothesis. For example, orf1p expression after viral clearance and ATRi or CHKi treatment to Huh7-J17 cells.

We have demonstrated that HCV impairs the HRR pathways and activates L1 retrotransposition. On the other hand, HCV also upregulated L1orf1p expression but we speculate its mechanism to be via DDR independent pathways. Since treating Huh7 cells with Chk1i does not influence L1orf1p level (data included as Supplementary Figure S5C). The possible mechanism of upregulation of L1orf1p by HCV is generation of stress granules where L1orf1p can get sequestered [42]. The exact causative relationship of these factors is not known and warrants further investigation. We have included this statement in the conclusion section.

8) Please provide the information of the mechanisms how HCV impairs the HRR pathway and how the HRR pathway regulates L1 retrotransposition and/or discuss the possibilities.

A possible explanation is included in the discussion: HCV encoded NS5a binds to RAD51AP1 and can impair the RAD51/RAD51AP1/UAF1 trimeric complex, thus impairing DNA repair and increasing sensitivity to DNA damaging agents [14]. Additionally, HR factors are known to restrict retrotransposition potentially by sterically blocking the formation of the second DNA nick during L1 retrotransposition, perhaps by recruiting ssDNA-coating proteins RPA1 and Rad51 [40]. (see lines 432-438)

Reviewer 2 Report

Remarks to the Author:

Dhondurao Sudhindar et al. report that HCV activates L1 expression in human liver tissues and L1 retrotransposition in a cell line using reporter systems. They suggest that this increase is mediated by dysregulation of the DNA-damage repair response. Also, they claim that the increased L1 retrotransposition persists in cells after viral clearance. The topic is interesting; however, some important controls are missing and the authors draw conclusions that are not supported by the data presented.

Below, I have noted some comments for the authors to consider when revising their paper for publication in Cancers.

Comments:

1-Several of the main conclusions of the paper are based on cell-based L1 retrotransposition assays performed in Huh7 and Huh7-J17. However, when the authors used the EGFP-based retrotransposition assay (fig 2D, S2 and S5) the number of green cells is too low in every condition to get any conclusion (i.e. in GFPLRE3 transfected cells 0.18% for Huh7 vs 0.60% for Huh7-J17). In fact, this small increase in absolute values could be due to differences in transfection efficiency (i.e. in EGFP transfected cells 42.71% for Huh7 and 46.09% for Huh7-J17). I consider that the authors should enrich the percentage of transfected cells, or not draw any conclusion based on this experiment. On the other hand, the blasticidin-based retrotransposition assay is more informative but require a viability control (i.e. transfection with pcDNA6.1) to avoid misinterpretation (Figure 2F and 5B).

2-The authors suggest that the HCV upregulates L1 retrotransposition by inhibition of DSB repair pathways. Does the endonuclease-independent LINE-1 retrotransposition occur in Huh7-J17 at higher levels than in Huh7? A retrotransposition assay using an Endonuclease- deficient LINE-1 should be performed to support this hypothesis.

3- Fig1A and Fig 1B. L1 is significantly higher expressed in the livers of CHC patients vs controls.  Do patients with CHC also have more L1 insertions? Do HCC tissues of CHC patients have more L1 insertions than HCC of other aetiology? 
4- Fig 1C. Has L1ORF1p expression also been analyzed in Pre-HCC and HCC of other aetiology? Comparing the number of positive and negative tissues of individuals in both circumstances is important to draw conclusions.
5- The title of the paper is too ambitious for the data presented. In fact, the influence of HCV on L1 retrotransposition beyond clearance is exclusively supported by a Blast-based retrotransposition experiment and the influence of HCV in cancer through L1 activation is speculative with the presented data.

Author Response

Dhondurao Sudhindar et al. report that HCV activates L1 expression in human liver tissues and L1 retrotransposition in a cell line using reporter systems. They suggest that this increase is mediated by dysregulation of the DNA-damage repair response. Also, they claim that the increased L1 retrotransposition persists in cells after viral clearance. The topic is interesting; however, some important controls are missing and the authors draw conclusions that are not supported by the data presented.

We wish to thank the reviewer for finding the study interesting and pointing out the missing controls (which are now included in the updated manuscript). Moreover, we have updated the title and conclusions as per the useful suggestions by the reviewer.

Below, I have noted some comments for the authors to consider when revising their paper for publication in Cancers.

See our point-by-point response below: 

Comments:

1-Several of the main conclusions of the paper are based on cell-based L1 retrotransposition assays performed in Huh7 and Huh7-J17. However, when the authors used the EGFP-based retrotransposition assay (fig 2D, S2 and S5) the number of green cells is too low in every condition to get any conclusion (i.e. in GFPLRE3 transfected cells 0.18% for Huh7 vs 0.60% for Huh7-J17). In fact, this small increase in absolute values could be due to differences in transfection efficiency (i.e. in EGFP transfected cells 42.71% for Huh7 and 46.09% for Huh7-J17). I consider that the authors should enrich the percentage of transfected cells, or not draw any conclusion based on this experiment. On the other hand, the blasticidin-based retrotransposition assay is more informative but require a viability control (i.e. transfection with pcDNA6.1) to avoid misinterpretation (Figure 2F and 5B).

We understand the concern raised by the reviewer regarding low number of cells exhibiting active retrotransposition, however the difference we have observed between Huh7 and Huh7-J17 is not down to difference in transfection efficiencies of the cell lines. Even after normalisation for the transfection efficiency there is an approximately 3-fold difference in the level of retrotransposition. Moreover, the EGFP-based RTN FACS data is fully supported by qPCR. The RTN plasmids have a puromycin resistance cassette that is normally utilised to enrich the transfected cells, however since Huh7-J17 cells are already resistant to puromycin due to the presence of the puromycin-resistance cassette integrated within the HCV replication construct we could not carry out the selection/enrichment of the transfected cells. For the blasticidin-based retrotransposition assay, cells transfected with pcDNA6.1-blast were indeed run alongside the RTN assay as controls. We have now included the images of the control plates in Figure 2F and 5B (this is 5C in the revised manuscript).

2-The authors suggest that the HCV upregulates L1 retrotransposition by inhibition of DSB repair pathways. Does the endonuclease-independent LINE-1 retrotransposition occur in Huh7-J17 at higher levels than in Huh7? A retrotransposition assay using an Endonuclease- deficient LINE-1 should be performed to support this hypothesis.

The suggested experiment might indeed support our hypothesis. However, the rate of endonuclease-independent LINE1 retrotransposition is much lower (more than 10-fold) than endonuclease-dependent LINE1 retrotransposition (Morrish et al., Nature Genetics 2002; Farkash et al., Nucleic Acids Res. 2006). Hence, we are afraid that it will be undetectable in our system. The experiment will need a lot of optimisation and is currently beyond the scope of the present study.

3- Fig1A and Fig 1B. L1 is significantly higher expressed in the livers of CHC patients vs controls.  Do patients with CHC also have more L1 insertions? Do HCC tissues of CHC patients have more L1 insertions than HCC of other aetiology? 

De novo L1 insertions in CHC patients without cancer have not been investigated yet and the results of our current study warrants this investigation in follow-on studies. Nevertheless, comparing L1 insertions in HCC of different aetiologies indicates a higher level of L1 insertions in virus-associated HCC versus alcohol-associated HCC. We had included the details in the ‘introduction section’ as: Shukla et al demonstrated L1 (an autonomous mobile genetic element, or retrotransposon) activation and their promotion of oncogenic signalling pathways in HBV and HCV related HCC [21]. Active retrotransposition has also been observed in cases of alcohol-related liver disease associated HCC [22], although the rate of somatic retrotransposition was lower in non-viral cases (8 somatic L1 insertions in 3 out of 25 individuals) compared to virus associated HCC (12 somatic L1 insertions in 5 out of 26 donors) [21,22].

Based on the reviewer’s query we have further analysed the retrotransposition data reported by the Pan Cancer Whole Genome Analysis study that included HCC cases. We have now integrated the data from different HCC retrotransposition studies and combined the number of cases evaluated so far for different aetiologies of HCC and updated the introduction as: Shukla et al demonstrated L1 (an autonomous mobile genetic element, or retrotransposon) activation and their promotion of oncogenic signalling pathways in HBV and HCV related HCC [21]. Active retrotransposition in virus (HCV or HBV) associated HCC was also evident from the Pan-Cancer Whole Genome Analysis (PCWGA) consortium study that included HCC cases [22]. Active retrotransposition has also been observed in cases of alcohol-related liver disease associated with HCC [22,23]. However, the rate of somatic retrotransposition was lower in alcohol-related cases (~14%; 5 out of 37 individuals) compared to virus associated HCC (~32%; 97 out of 306 individuals) [21-23]. (see lines 86-95)

4- Fig 1C. Has L1ORF1p expression also been analyzed in Pre-HCC and HCC of other aetiology? Comparing the number of positive and negative tissues of individuals in both circumstances is important to draw conclusions.

We would like to thank the reviewer for raising this query. As suggested by the reviewer, we have analysed L1orf1p expression by IHC for 5 pre-HCC liver biopsies of patients with fatty liver disease associated with alcohol and metabolic syndrome from our biobank. 1 out of 5 (20%) were found to be positive. In addition, we interrogated the TCGA-LIHC RNAseq dataset for L1 expression in the non-tumour liver of HCC patients with alcohol related aetiology.  A trend of increase in L1 transcripts in the non-tumour tissue of patients with alcohol related HCC was observed but this did not reach statistical significance when compared to individuals with no history of any known HCC risk factors. The data is included as Supplementary Figure S1 in the revised manuscript. New L1 IHC data is also included in the results (see line 249-255) and accordingly the title and conclusion of this section has been modified to account for the new results.

We understand that the sample size is too small to actually carryout a direct comparison between different aetiologies however, liver biopsy procedure carries significant risks (haemorrhage, tumour seeding) and thus is mostly avoided. These are the only pre-HCC cases available in our biobank. To increase sample size, a prospective collection needs to be done or samples from other sites needs to be procured, which is out of scope of current study. We plan to obtain future funding to carry out this follow on work.

5- The title of the paper is too ambitious for the data presented. In fact, the influence of HCV on L1 retrotransposition beyond clearance is exclusively supported by a Blast-based retrotransposition experiment and the influence of HCV in cancer through L1 activation is speculative with the presented data.

We have taken the comment into consideration and have modified the title as follows:

HCV Activates Somatic L1 Retrotransposition – a potential hepatocarcinogenesis pathway